# In Vivo Confirmation of the Antimicrobial Effect of Probiotic Candidates against *Gardnerella vaginalis*

**DOI:** 10.3390/microorganisms9081690

**Published:** 2021-08-09

**Authors:** Hyemin Kim, YongGyeong Kim, Chang-Ho Kang

**Affiliations:** MEDIOGEN, Co., Ltd., Biovalley 1-ro, Jecheon-si 27159, Korea; astrongirl.hm@gmail.com (H.K.); yongkyung@naver.com (Y.K.)

**Keywords:** bacterial vaginosis, *Gardnerella vaginalis*, probiotics, lactic acid

## Abstract

Bacterial vaginosis (BV) is caused by a microbial imbalance of the vaginal ecosystem, causing genital discomfort and potentially even various complications in women. Moreover, research on the treatment or prevention of BV is increasing. In this study, we evaluated the antimicrobial and anti-inflammation effects of the lactic acid bacteria (LAB) *Ligilactobacillus salivarius* MG242, *Limosilactobacillus fermentum* MG901, and *Lactiplantibacillus plantarum* MG989 in a BV-induced mice model. The oral administration of the LAB significantly inhibited the growth of *Gardnerella vaginalis* up to 43% (*p* < 0.05). The LAB downregulated the expression of pro-inflammatory cytokines (IL-1β and TNF-α) and myeloperoxidase (*p* < 0.05). Upon histological examination, the exfoliation of epithelial cells in the vaginal tissues was found to be reduced in the probiotic administration group compared to the infected group. In addition, the LAB tolerated the gastric and/or intestinal simulated conditions and proliferated, showing potential in promoting health based on hemolysis activity, antibiotic susceptibility, enzyme activity, and lactic acid production. Altogether, our results showed that the investigated LAB may be a good food ingredient candidate for ameliorating BV in women.

## 1. Introduction

Bacterial vaginosis (BV) is the most common vaginal infection in women of childbearing age. BV causes genital discomfort (exudate, pruritus, dyspareunia, or malodorous discharge) or complications (pelvic inflammatory disease, spontaneous abortion, preterm birth, and infections of the post-operative wound, among others) [1,2]. BV is caused by dysbiosis in the vaginal flora; the vaginal microbiota in healthy women is mainly composed (up to 90%) of lactic acid bacteria (LAB) [3]. However, the overgrowth of certain pathogenic bacterial genera, such as *Gardnerella*, *Prevotella*, *Megasphaera*, *Atopobium*, or *Dialister*, causes vaginal inflammatory disease in the genital mucosa [3,4]. Some strains of *G. vaginalis* are a major contributor to bacterial vaginosis due to its strong adherence to vaginal cells and its biofilm-forming capacity, which serves as a platform for the attachment of other BV-related species [5,6]. Therefore, suppressing the proliferation of *G. vaginalis* is a possible strategy for the treatment of BV. The currently available pharmaceuticals for BV rely on antibiotics, such as metronidazole or clindamycin [6,7]. Although antibiotics can temporarily relieve BV symptoms, the use of antibiotics not only poses an antibiotic resistance problem, but also kills a wide range of beneficial vaginal microbes [8,9,10]. Thus, new fundamental treatment or prevention strategies of BV are required.

Probiotics could be an alternative to antibiotics. According to the widely recognized FAO/WHO definition, revised by Hill et al., probiotics are living microorganisms that confer health benefits to the host when administered in adequate amounts [11]. Probiotics provide various curative effects in the form of immune system stimulation, intestinal microflora balance, and anticarcinogenic properties, among others [12]. The probiotics ingested through the oral cavity can be discharged through the stomach and intestines and naturally move to the vaginal entrance [13]. Probiotics, especially LAB, are considered to have beneficial effects in the vagina via four main mechanisms: (1) Antimicrobial activity via compounds such as lactic acid, hydrogen peroxide (H_2_O_2_), or bacteriocins; (2) inhibition of pathogenic biofilms; (3) co-aggregation with pathogens; and (4) modulation of the immune response [2,14]. It has been observed from previous studies of probiotics for BV that the oral administration of *Lactobacillus acidophilus* GLA14 and *Lacticaseibacillus rhamnosus* HN001 can attenuate experimentally induced BV in mice [15]. *Lactiplantibacillus plantarum* NK3 and *Bifidobacterium longum* NK49 were shown to increase the levels of TNF-α and myeloperoxidase (MPO) in the mice vagina and uterus and decrease the IL-10 level in the uterus [16]. Preliminary studies of our group have also revealed that the LAB strains *Ligilactobacillus salivarius* MG242, *Limosilactobacillus fermentum* MG901, and *Lactiplantibacillus plantarum* MG989, which were isolated from the vaginas of Korean women, show antipathogenic activity against *G. vaginalis* or *Candida albicans*, while the mixture of strains shows a synergistic effect [17,18,19,20]. However, these studies were unable to provide evidence of the antimicrobial effect of the probiotic candidates against BV.

The purpose of this study was to establish whether probiotics, which could be helpful for gut health, could also help to maintain women’s vaginal health. The present study identified the in vitro and in vivo efficacy of the LAB strains in ameliorating BV symptoms in a G. vaginalis infection-induced BV mouse model. In addition, the probiotic safety was confirmed in order to apply in functional foods. Therefore, it is expected that BV can be prevented through the intake of probiotic health-related functional food.

## 2. Materials and Methods

### 2.1. Cultivation

The LAB strains *Ligilactobacillus salivarius* MG242, *Limosilactobacillus fermentum* MG901, and *Lactiplantibacillus plantarum* MG989 used in this study were supplied by MEDIOGEN Co., Ltd. (Jecheon, Korea). These strains were isolated from a healthy woman’s vagina. The LAB were activated by culturing them in de Man, Rogosa and Sharpe (MRS, Difco, Detroit, MI, USA) broth at 37 °C for 15 h.

*Gardnerella vaginalis* (GV) KCTC5096 was obtained from the Korean Collection for Type Cultures (KCTC, Daejeon, Korea). The GV strain was cultured in a modified brain–heart infusion (mBHI; Difco, Detroit, MI, USA) broth containing 10% horse serum (Life Technologies Co., Grand Island, NY, USA), 1% yeast extract (Difco, Detroit, MI, USA), 0.1% maltose, and 0.1% glucose, and cultivated at 37°C for 48 h under anaerobic conditions (BD GasPak^TM^ EZ pouch systems, BS, USA). The cells were harvested and suspended in sterilized phosphate-buffered saline (PBS; pH 7.0) at a density of 5 × 10^6^ colony forming units (CFU)/mL for vaginal injection.

### 2.2. Antimicrobial Effect of the LAB Metabolites against the Growth of GV

The LAB were cultured in MRS broth for 15 h at 37 °C under aerobic conditions. The cultures were standardized to approximately 7 log CFU/mL of MRS and incubated for an additional 20 h at 37 °C under in an atmosphere of 10% CO_2_ without shaking. The cultures were centrifuged at 4000× *g* for 5 min at 4 °C, after which they were filtered using a 0.2 μm cellulose acetate membrane filter (ADVANTEC, Tokyo, Japan) to obtain cell-free supernatant (CFS).

To measure the anti-*G. vaginalis* activity, 1 × 10^6^ CFU *G. vaginalis* was inoculated in fresh mBHI media containing 10% of the CFS and co-cultured anaerobically at 37 °C for 36 h. The viable cell count of *G. vaginalis* was measured by diluting and plating on BHI agar with 5% horse blood (MB cell, Seoul, Korea) for 36 h with 12 h intervals. After inoculation for 24 h at 37 °C under anaerobic conditions, the CFUs were counted.

### 2.3. BV-Induced Mice Model and Administration of the LAB

Seven-week-old 57BL/6J female mice (weighing 19–22 g) were supplied by OrientBio Co. (Seongnam, Korea). The mice were housed in wire cages under climate-controlled conditions (50% ± 10% humidity and 20–24 °C), fed standard laboratory chow, and allowed water ad libitum. The animal experiments were approved by the OSONG Medical Innovation Foundation Institutional Animal Care and Use Committee (No. KBIO-IACUC-2020-072).

For the induction of BV by *G. vaginalis* infection, mice were randomly distributed into five groups (*n* = 5/group): Normal (NOR), control (CON), MG242, 1:1 mixture of MG901 and MG989 (M2), and 1:1:1 mixture of MG242, MG901, and MG989 (M3). In a previous study, MG901 and MG989 showed a synergistic effect in inhibiting *Candida albicans* [20]. Thus, we set the LAB treatment groups as MG242, M2, and M3. All mice, except for those of the normal group, were injected intraperitoneally with 1 mg of β-estradiol in 200 mL of filter-sterilized olive oil on the day of *G. vaginalis* inoculation. *G. vaginalis* was inoculated twice, on days 7 and 14 after the mice were obtained. The mice were anesthetized with isoflurane and inoculated vaginally with 5 × 10^6^ CFUs of *G. vaginalis* in 20 μL of sterile PBS (pH 7.0). The NOR group was treated with PBS instead of with *G. vaginalis*.

To prepare the probiotic samples, freshly harvested bacterial pellets (resulting from centrifugation at 4000× *g* for 10 min at 4 °C) were mixed with a cryoprotectant mixture [21] and freeze-dried. The dried cells were powdered, sealed to keep moisture out, and stored at 4 °C until further use. The probiotic powder (3 × 10^10^ CFUs/g of MG242, 1 × 10^11^ CFUs/g of MG901, and 1× 10^11^ CFUs/g of MG989) was prepared in deionized water (DW) and orally administered (5 × 10^9^ CFU/300 μL/head) once a day for 7 days, beginning 7 days after the first inoculation using *G. vaginalis*. The NOR and CON groups were administered DW instead of probiotics. On the day after the last administration of probiotics, the mice were sacrificed, and the excised vaginas were washed with 200 μL of PBS (pH 7.0). After washing, the vaginas were stored at –80 °C for mRNA extraction or histological examination. Serial diluted vaginal wash fluid (200 μL) was added to *G. vaginalis*-selected agar (Columbia blood agar with 10% horse blood, GV selective supplement; Oxoid, Basingstoke, United Kingdom) to determine the number of *G. vaginalis* CFUs, and the *G. vaginalis* inhibition rate was calculated using the following formula (Equation (1)):Inhibition rate (%) = (A − B)/A × 100(1)
where A refers to the log CFUs/mL of *G.vaginalis* in the control group and B refers to the log CFUs/mL of *G.vaginalis* in test group.

### 2.4. RNA Extraction and RT-PCR

The total mRNA was extracted manually from the vaginal tissues using an RNeasy Mini Kit (Qiagen, Hilden, Germany) and RT-PCR was performed using a TOP realTM One-Step PT-qPCR Kit (Enzynomics Co., Ltd., Daejeon, Korea). The following primers were used for the analysis: Interleukin-1β (IL-1β) forward: 5′-CAAGGAGAACCAAGCAACGA-3′, reverse: 5′-GGGTGTGCCGTCTTTCATTA-3′; tumor necrosis factor-α (TNF-α) forward: 5′-CTGTAGCCCACGTCGTAGC-3′, reverse: 5′-TTGAGATCCATGCCGTTG-3′; MPO forward: 5′-GAGTCCCACTCAGCAAGGTC-3′, reverse: 5′-TCTGGCGATTCAGTTTGGCT-3′; and β-actin forward: 5′-CAGCCTTCCTTCTTGGGTATG-3′, reverse: 5′-GGCATAGAGGTCTTTACGGATG-3′. The thermal cycling conditions were as follows: 50 °C for 30 min, 95 °C for 15 min, followed by 45 cycles of denaturation and then amplification at 95 °C for 5 s and 63 °C for 30 s. The relative quantity of the target mRNA (IL-1β, TNF-α, and MPO) was determined using the comparative CT method via normalization to the values of β-actin, which was used as a housekeeping gene.

### 2.5. Histopathological Analysis

The effect of the probiotic candidates on *G. vaginalis*-infected mice was evaluated by analyzing the histopathological changes in the vaginal tissues (EBO Co. Ltd., Cheongju, Korea). The vaginal tissues were fixed in 10% formalin for at least 24 h, dehydrated, embedded in paraffin, sectioned into 4 μm slices, and stained with hematoxylin and eosin. The stained slices were permounted (Fisher Scientific, Fair Lawn, NJ, USA) and subjected to microscopic examination (CKX41, Olympus Inc., Tokyo, Japan).

### 2.6. Adhesion

The ability of the LAB strains to adhere to HeLa cells was measured according to Joo et al. [22]. HeLa cells were cultured in RPMI 1640 (Gibco, Grand Island, NY, USA) supplemented with 10% DFBS (Gibco, Grand Island, NY, USA), and incubated in an atmosphere of 5% CO_2_ at 37 °C for 2 days. The HeLa cell suspension was seeded at a density of 5 × 10^4^ cells/mL in 12-well plates. Cultured HeLa cells were washed twice with sterile PBS. Each strain suspension (1 mL, 2 × 10^8^ CFUs/mL) was added to each well, followed by incubation in 10% CO_2_ for 1 h at 37 °C. After incubation, the cells were washed three times with sterile PBS and lysed. To determine the viable cell number of the LAB strains, the dilutions were plated on BHI agar with 5% horse blood (MB cell, Seoul, Korea) and the bacterial colony number was determined. 

### 2.7. Antibiotic Susceptibility

The antibiotic susceptibilities of three probiotic candidates were assayed using the minimum inhibitory concentration (MIC) test strip method. Nine antibiotic strips were used for testing the bacterial strains, namely, ampicillin, chloramphenicol, clindamycin, erythromycin, gentamicin, kanamycin, streptomycin, tetracycline, and vancomycin (Liofilchem, Abruzzi, Italy). The bacteria were grown for 18 h at 37 °C in MRS medium. The cells were harvested via centrifugation at 3470× *g* for 5 min, washed twice with PBS (pH 7.0), and resuspended in PBS to a McFarland turbidity of 0.5. The cell suspensions were inoculated on BHI agar using swabs. The plates were dried for 15 min, and the MIC test strips were placed on the agar surface according to the manufacturer’s instructions. The plates were then incubated at 37 °C, and the results were assessed after 20 h of inoculation, according to the European Food Safety Authority (EFSA) guidelines [23]. 

### 2.8. Assessment of Enzyme Production

To measure the enzyme activity, each LAB strain was grown on an MRS agar plate for 18 h at 37 °C. Each of the strains was assayed using an API ZYM system with cell colonies, according to the manufacturer’s instructions (bioMérieux, Marcy L’Ètoile, France). The enzyme activity was determined according to the intensity of coloration. 

### 2.9. Analysis of the Lactic Acid Level Using the HPLC-UV Method

The total lactic acid production in the culture supernatant was analyzed using HPLC-UV with a Chiralpak^®^ MA (+) column (reverse phase-type, 4.6 × 50 mm, 5 μm; Daicel Chemicals Industries Ltd., Tokyo, Japan). The sample injection volume was 10 μL. The mobile phase contained 2 mM CuSO_4_ and was eluted at a flow rate of 1.0 mL/min. The effluent was monitored at 254 nm using a UV detector. L-(+)-lactic acid and D-(-)-lactic acid solutions were used as standard solutions.

### 2.10. Hemolysis Activity

For evaluating the hemolytic activity, the selected strains were grown in MRS for 18 h at 37 °C, streaked onto tryptic soy agar (Difco, Detroit, MI, USA) medium with 5% sheep blood (MB cell, Seoul, Korea), and incubated at 37 °C for 48 h. The controls for *β*-hemolysis and *α*-hemolysis used *Enterococcus faecalis* AHC1 and *Lactobacillus salivarius* MG4265, respectively.

### 2.11. Bile Salt Hydrolase Activity

The bile salt hydrolase (BSH) activity was determined as described by Sheheta et al. [24]. The LAB were grown on MRS agar plates containing 0.5% (*w*/*v*) taurodeoxycholic acid sodium salt (TDCA; Sigma, St, Louis, MO, USA) and 0.037% calcium chloride. The plates were incubated under anaerobic conditions at 37 °C for 72 h. The precipitation zone surrounding the colonies indicated the BSH activity of the bacteria.

### 2.12. Statistical Analysis

For the antimicrobial effects of the LAB metabolites against the growth of *G. vaginalis*, all data are presented as the mean ± standard deviation (SD) of the mean (*n* = 3). For in vivo experiments, all data are presented as the mean ± standard error (SE) of the mean (*n* = 5). The analysis was conducted using SPSS statistics software (IBM; Armonk, NY, USA). Statistical significance was analyzed using one-way analysis of variance, followed by post-hoc analysis using Dunnett’s comparison tests.

## 3. Results

### 3.1. Antimicrobial Effect of the LAB Metabolites against the Growth of GV

To investigate the antimicrobial effect from the metabolites produced by the LAB, the growth inhibitory activity of the CFS of the LAB against GV was measured (Table 1). This experiment was conducted to confirm the antibacterial effect of the metabolites of live lactic acid bacteria or not. The growth of GV increased during the 24 h of exposure time; however, the number of GV colonies decreased after 36 h of exposure time (*p* < 0.05). Among the LAB, MG242 showed a relatively higher inhibition of the growth of GV than other strains. 

### 3.2. In Vivo Antimicrobial Effect of the LAB on BV-Induced Mice

The inhibitory effect of the LAB on *G. vaginalis*-infected mice was observed (Figure 1). The BV-induced group (CON) showed the highest cell counts (7.41 ± 0.17 log CFUs/mL) of *G. vaginalis*. Moreover, all of the LAB treatment groups reduced the number of *G. vaginalis* colonies as showing MG242 (6.62 ± 0.72 CFUs/mL), M2 (6.74 ± 0.73 CFUs/mL), and M3 (4.26 ± 0.72 CFUs/mL). Among the LAB, M3 significantly reduced the *G. vaginalis* cell number up to 43% against the control (*p* < 0.05).

### 3.3. Pro-Inflammatory Biomarkers of the Vaginal Tissues in BV-Induced Mice

The mRNA levels of the pro-inflammatory factors were measured in the vaginal tissues of BV-induced mice after the LAB administration as a biochemical index reflecting the degree of neutrophil infiltration (Figure 2). The BV-induced vaginas were accompanied by the upregulation of the mRNA levels of all measured cytokines and an MPO activity increase (*p* < 0.05). On the contrary, the oral administration of the LAB downregulated the pro-inflammatory factor levels. Among the LAB, M2 significantly inhibited the activity of IL-1β, TNF-α, and MPO (*p* < 0.05; Figure 2).

### 3.4. Histopathological Analysis of BV-Induced Mice

The extracted vaginal tissues were washed and fixed, followed by staining with hematoxylin and eosin (H&E) to confirm the changes in the vaginal tissues (Figure 3). Compared with that of the NOR group, the vaginal epithelial tissue of the mice in the CON group thickened and appeared to have inflammatory cell infiltration. In addition, histological examination revealed that the exfoliation of epithelial cells in the vaginal tissues due to *G. vaginalis* infection was significantly increased in the CON group. Compared with that in the CON group, the exfoliation in the M2 and M3 groups was significantly reduced in the mice of the LAB treatment group.

### 3.5. Fulfillment of the LAB

We investigated the ability of the LAB to adhere to human cervical epithelial (HeLa) cells. Each LAB strain showed an adhesive ability of 7.03 ± 0.01 to 7.32 ± 0.08 log CFUs/wall (data not shown). On the contrary, the strain mixture condition M3 presented the highest ability to adhere to HeLa cells (7.71 ± 0.01 log CFUs/wall, *p* < 0.05).

The antibiotic resistance of the LAB was assessed using an MIC test. The results of the LAB were within the epidemiological cut-off values suggested by the EFSA [23]. Although *Lig. salivarius* MG242 showed no antibiotic resistance to the eight antibiotics tested, *Lim. fermentum* MG901 and *Lac. plantarum* MG989 showed antibiotic resistance to chloramphenicol (Table 2).

The enzymatic activity patterns of the LAB were assessed using an API ZYM system (Table 3). The probiotic candidates must be evaluated for the production of appropriate enzymes in order to avoid the production of potentially toxic substances. The selected strains did not produce lipase, *β*-glucuronidase, *N*-acetyl-β-glucosaminidase, or α-mannosidase. Among them, β-glucuronidase is a bacterial carcinogenic enzyme that exerts negative effects on the liver [25].

Lactic acid production was measured using the HPLC-UV method. All strains could produce D- and L-lactate (Table 4). Overall, the LAB produced a higher content of L-(**+**)- lactic acid than D-(**-**)-lactic acid. The highest D- (**-**)-lactate production was achieved by *Lim. fermentum* MG901 (40.7%).

The probiotic candidates showed neither alpha nor beta hemolysis on the blood agar plates. No hemolytic activity was detected in any of the strains tested (MG242, MG901, or MG989) (data not shown). Moreover, the plate assay for the BSH activity showed that all strains were negligibly positive or negative for bile salt hydrolase (data not shown).

## 4. Discussion

The present study investigated the beneficial advantages of LAB in a BV-induced mice model and demonstrated that the strains ameliorated BV. The LAB showed antimicrobial efficacy and reduced the vaginal epithelial tissue exfoliation caused by G. vaginalis infection in vitro or in vivo. Interestingly, our results revealed that the LAB treatment groups inhibited the growth of G. vaginalis up to 43% within a week. Further study will be needed to investigate the antimicrobial effect along the long-term intake. The antimicrobial efficacy might be the effect of the lactic acid produced by the LAB. Although the beneficial vaginal microbiota releases various antimicrobial substances, recent studies have revealed that the main anti-pathogen factor is the lactic acid produced by the LAB, which leads to vaginal eubiosis [26,27,28]. According to Hemalatha et al. [29], the mean vaginal pH in women with BV is higher than that in women without BV (pH 4.6). Moreover, to maximize the antibacterial properties of lactic acid, LAB need to acidify the vagina to a pH of 3.9 [28]. Thus, the LAB used in our study could prevent or relive BV by creating an acidic vaginal environment and by showing a high adhesive ability, inhibiting the occurrence or growth of G. vaginalis.

The overgrowth of G. vaginalis caused vaginal inflammation through the upregulation of IL-1β, TNF-α, or MPO in vivo. However, our LAB showed immunomodulation, as shown by the downregulation of the activity of IL-1β, TNF-α, and MPO. Although there were no significant differences within the LAB treatments, M2 exhibited a synergistic effect to reduce the levels of pro-inflammatory cytokines. This might have occurred through the production of lactic acid by MG901, which produced D-(**-**)-lactic acid at an almost equal ratio as that of L-(**+**)-lactic acid (Table 4). Lactic acid is the primary molecule responsible for the acidification of the vagina and is known to have antimicrobial and immunomodulatory functions [28,30,31]. LAB produce D-(**-**)-/ L-(**+**)-lactic acid, or both D-(**-**)- and L-(**+**)-lactic acid, and the production ratio of the lactic acid types is different for each strain [32]. According to Witkin et al. [33], D-(**-**)-lactic acid could contribute to GV suppression and immunity: The higher protection provided by L. crispatus, compared to that provided by L. iners against uropathogens and the associated adverse pregnancy outcomes has been attributed to the greater protective role of D-(**-**)-lactic acid, compared to that of the L-isomer [4,34,35].

Recently, probiotics are being applied not only as food or dietary supplements, but also in preclinical and clinical trials [36]. To identify their health benefits, in addition to a safety assessment, the functionality and the technical specifications of probiotic candidates should be demonstrated. The important properties of probiotics as functional supplements are their acid tolerance and their adhesive ability to survive and proliferate under gastrointestinal conditions [37,38,39]. Our LAB strains are likely to survive in both the stomach and intestinal environments, as they have shown high survivability in various acidic conditions [17,18]. The tolerance to acidic environments of the LAB has been reported to be associated to changes in their glycolytic flux, their ability to control the intracellular pH, or their cell membrane ATPase [40]. Their adhesion to epithelial cells is also an important probiotic function for pathogen colonization prevention. Biofilm formation is one of the most important virulence factors, causing increased toxicity, as well as the antibiotic or antimicrobial byproducts produced by the attached vaginal flora [41,42]. G. vaginalis has a key role in the constitution of the biofilm in vaginal epithelial cells and causes gynecological infections [5,43]. Thus, the competition for vaginal cell adhesion between probiotics and G. vaginalis is considered a beneficial effect for inhibiting the growth of pathogens [44].

The use of probiotics with antibiotic resistance has caused concerns due to the transmission of their antibiotic resistance genes to pathogens through horizontal gene transfer. In our LAB strains, Lim. fermentum MG901 and Lac. plantarum MG989 showed antibiotic resistance to chloramphenicol (Table 3). When probiotic strains are killed by antibiotics ingested for therapeutic purposes, their functionality is decreased. In this respect, the antibiotic resistance of probiotic microorganisms is thought to be advantageous for survival in the gastrointestinal tract during medical treatment with antibiotics [45]. Therefore, resistance to antibiotics is also recognized as a very important factor to consider [46]. Likewise, probiotics have a double-edged sword effect: They can be useful for individuals with unbalanced intestinal microflora due to the administration of various antibiotics, but they may also transfer resistance genes to other bacteria [47]. Consequently, the U.S. FDA evaluated the “safety of probiotic use” with “reasonable certainty” [47].

In conclusion, we demonstrated the beneficial effect of Ligilactobacillus salivarius MG242, Limosilactobacillus fermentum MG901, and Lactiplantibacillus plantarum MG989 for BV. Although each strain represented antimicrobial and anti-inflammatory effects, the mixture of the LAB showed synergistic effects in reducing the number of G. vaginalis colonies and the inflammatory cytokine levels (TNF-a, IL-1b, and the MPO activity) in the vaginas of BV-induced mice. Our results also showed that these LAB fulfilled the requirements of probiotics, showing stability (acid tolerance and adhesion) and safety (absence of hemolytic, β-glucuronidase, and bile salt hydrolase activity), which means that they can be applied to functional and nutraceutical dietary supplements for BV. However, more safety evaluation tests are required before the employment of these LAB strains for the production of functional food products. Moreover, further clinical trials are required to evaluate the efficacy of these LAB on BV.

## Figures and Tables

**Figure 1 microorganisms-09-01690-f001:**
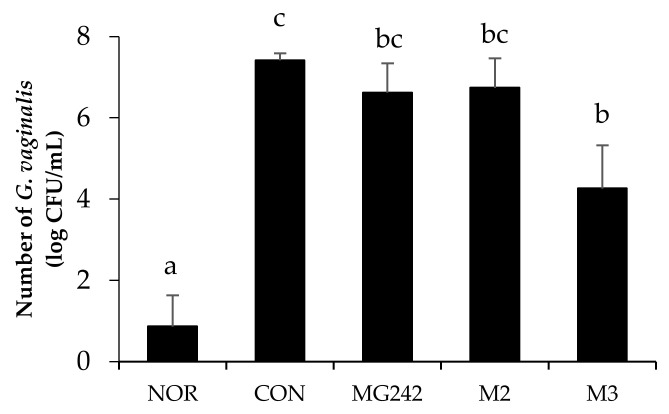
Antagonistic effect of the LAB against the growth of *Gardnerella vaginalis*. Data expressed as the mean ± standard error (SE). Statical difference among groups was analyzed using Tukey’s multiple comparison tests, and different letters (a, b, bc, and c) indicate a significant difference at *p* < 0.05. NOR, normal; CON, control; MG242, *Lig.*
*salivarius* MG242; M2, 1:1 mixture of *Lim. fermentum* MG901 and *Lac. plantarum* MG989; M3, 1:1:1 mixture of *Lig.*
*salivarius* MG242, *Lim. fermentum* MG901, and *Lac. plantarum* MG989.

**Figure 2 microorganisms-09-01690-f002:**
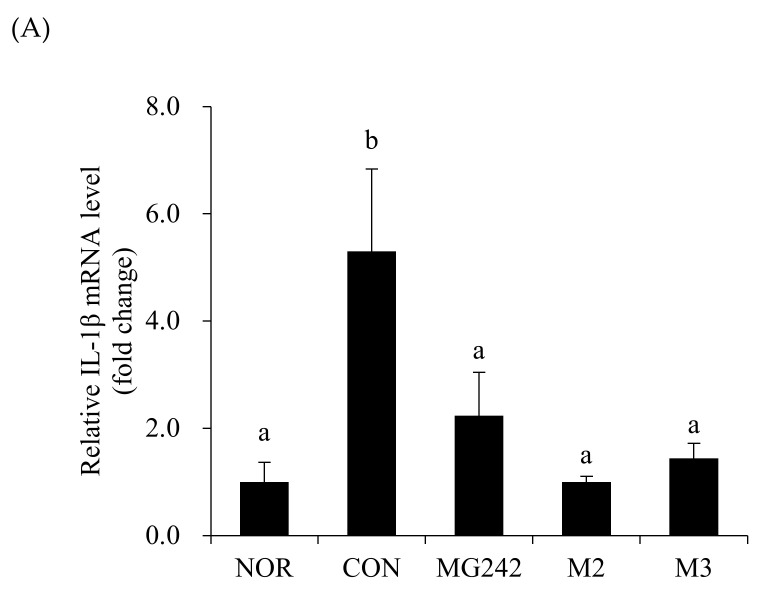
Pro-inflammatory biomarkers in BV-induced mice. RT-PCR analysis of the mRNA expression of (**A**) IL-1β, (**B**) TNF-α, and (**C**) myeloperoxidase (MPO). Glyceraldehyde-3-phosphate dehydrogenase (GAPDH) was used as a housekeeping gene to normalize all samples. Data are expressed as the mean ± SE. The statistical differences among groups were analyzed using Tukey’s multiple comparison tests, and the different letters (a and b) indicate a significant difference at *p* < 0.05. NOR, normal; CON, control; MG242, *Lig.*
*salivarius* MG242; M2, 1:1 mixture of *Lim. fermentum* MG901 and *Lac. plantarum* MG989; M3, 1:1:1 mixture of *Lig.*
*salivarius* MG242, *Lim. fermentum* MG901, and *Lac. plantarum* MG989.

**Figure 3 microorganisms-09-01690-f003:**
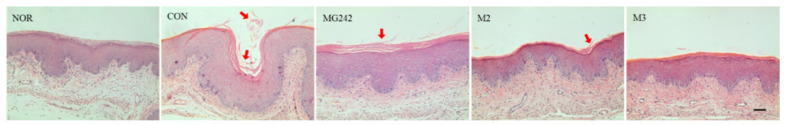
Histological changes in the vaginal tissues of BV-induced mice. Representative photographs of the hematoxylin and eosin (H&E)-stained vaginal tissue sections (200× magnification, scale bar = 50 μm). Arrows indicate exfoliated epithelial cells. NOR, normal; CON, control; MG242, *Lig.*
*salivarius* MG242; M2, 1:1 mixture of *Lim. fermentum* MG901 and *Lac. plantarum* MG989; M3, 1:1:1 mixture of MG242, MG901, and MG989.

**Table 1 microorganisms-09-01690-t001:** Characterization of the antimicrobial properties of the LAB strains.

Treatment	Exposure Time (log CFU/mL)			
	T_0_	T_12_	T_24_	T_36_
Control (mBHI with 10% MRS broth)	6.93 ± 0.014	7.69 ± 0.004 ^a^	8.07 ± 0.015 ^a^	6.56 ± 0.021 ^a^
*Lig. salivarius* MG242		7.28 ± 0.013 ^d^ (61.72%)	7.90 ± 0.028 ^c^ (33.05%)	6.17 ± 0.012 ^c^ (59.78%)
M2 (MG901 + MG989)		7.39 ± 0.014 ^c^ (49.90%)	7.97 ± 0.028 ^b^ (21.19%)	6.29 ± 0.009 ^b^ (16.10%)
M3 (MG242 + MG901 + MG989)		7.54 ± 0.010 ^b^ (30.51%)	7.94 ± 0.003 ^bc^ (26.69%)	6.12 ± 0.043 ^c^ (64.16%)

The results are expressed as the mean ± standard deviation (SD); each data point represents the average of three repeated measurements from three independently replicated experiments. The statistical difference among groups was analyzed using Tukey’s multiple comparison tests, and the different letters (a, b, bc, c, and d) indicate a significant difference at *p* < 0.05. Inhibition rates from the controls are represented in parentheses. M2, 1:1 mixture of *Lim. fermentum* MG901 and *Lac. plantarum* MG989; M3, 1:1:1 mixture of *Lig.*
*salivarius* MG242, *Lim. fermentum* MG901, and *Lac. plantarum* MG989.

**Table 2 microorganisms-09-01690-t002:** Minimum inhibitory concentration test results for the LAB.

Antibiotics (μL/mL)	*Lig.**salivarius* MG242		*Lim. fermentum* MG901		*Lac. plantarum* MG989	
	MIC	EFSA	MIC	EFSA	MIC	EFSA
Ampicillin	0.094	4	0.038	2	0.75	2
Gentamycin	2	16	0.125	16	0.094	16
Kanamycin	64	64	2	64	24	64
Streptomycin	24	64	1.5	64	n.r.	n.r.
Tetracycline	0.75	8	6	8	32	32
Chloramphenicol	4	4	12	4	12	8
Erythromycin	0.047	1	0.023	1	0.19	1
Clindamycin	0.064	4	<0.016	4	0.094	4

n.r., not required; MIC, minimum inhibitory concentration; EFSA, EFSA cut-off value [23]; *Lig, Ligilactobacillus*; *Lim, Limosilactobacillus*; *Lac, Lactiplantibacillus*.

**Table 3 microorganisms-09-01690-t003:** Enzymatic activity of the LAB, as measured using an API ZYM system.

Enzyme	*Lig. Salivarius* MG242	*Lim. Fermentum* MG901	*Lac. Plantarum* MG989
Alkaline phosphatase	1	0	1
Esterase (C4)	1	3	1
Esterase lipase (C8)	1	2	1
Leucine arylamidase	4	3	4
Valine arylamidase	1	1	3
Cystine arylamidase	1	1	1
Acid phosphatase	3	1	2
Naphthol-AS-BI-phosphohydrolase	3	2	2
α-Galactosidase	1	4	0
β-Galactosidase	0	5	5
α-Glucosidase	0	3	3
β-Glucosidase	0	0	4
*N*-Acetyl-β-glucosaminidase	0	0	5

All strains were negative for lipase (C14), trypsin, α-chymotrypsin, β-glucuronidase, α-mammnosidase, α-fucosidase; *Lig, Ligilactobacillus*; *Lim, Limosilactobacillus*; *Lac, Lactiplantibacillus*.

**Table 4 microorganisms-09-01690-t004:** Comparison of the lactic acid isomers produced by lactic acid bacterial strains.

Strains	Lactic Acid Content (g/L)			Isomer Ratio (%)	
	D (-)	L (+)	Total (D + L)	D (-)	L (+)
*Lig. salivarius* MG242	1.1	16.4	17.4	6.3	94.3
*Lim. fermentum* MG901	4.8	7.0	11.8	40.7	59.3
*Lac. plantarum* MG989	0.9	10.9	11.8	7.6	92.4

*Lig, Ligilactobacillus; Lim, Limosilactobacillus; Lac, Lactiplantibacillus.*

## Data Availability

The data presented in this study are available on request from the corresponding author.

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
