# Peer review of "In Vivo Confirmation of the Antimicrobial Effect of Probiotic Candidates against Gardnerella vaginalis"

_microorganisms, 2021, doi:10.3390/microorganisms9081690_

Round 1

Reviewer 1 Report

In this manuscript, Kim et al. investigated the antimicrobial and anti-inflammatory effects of LAB strains in a BV-induced mice model. Generally, the topic of the present work could be current but the finalization doesn’t extend the current knowledge in the field of probiotics and their application. The review of literature, statement of the problem and framing of the study aims are not sufficiently clear and exhaustive. The organization of materials and methods section is rather confusing, some fundamental information are missing, such as the number of mice inserted in each experimental group. It is not clear why the antimicrobial effect was evaluated in vitro using the supernatant of LAB cultures, when instead the experimentation on mice was carried out with lyophilized bacteria. Furthermore, it would have been useful to include in paragraph 2.3 also the evaluation, at the end of the treatment, of the presence of the administered probiotics in the vaginal samples. This data would have confirmed the passage of bacterial strains in the vagina and their ability to adhere.

The opinion of the referee is that, despite the undoubted interest in the topic addressed by the authors, the work needs greater clarity in planning and goal setting.

Author Response

Point 1. Generally, the topic of the present work could be current but the finalization doesn’t extend the current knowledge in the field of probiotics and their application.

Response: We are planning to develop a functional probiotic product for women’s vaginal health based on the present study. And Human clinical trials are now established. Thus, this study is a baby step for application to functional probiotics product. Our further study would be provided the extended knowledge. Thank you.

Point 2. The review of literature, statement of the problem and framing of the study aims are not sufficiently clear and exhaustive.

Response: We revised the introduction part and clarified our goal in revised MS (Line 66-72). Your comments helped make this study more convincing. Thank you.

Point 3. The organization of materials and methods section is rather confusing, some fundamental information are missing, such as the number of mice inserted in each experimental group.

Response: We added the number of mice per group in revised MS (Line 120).

As mentioned in introduction part, these three strains were identified as probiotics in the previous study. However, we found some missing data for the LAB about safety (Antibiotic susceptibility, enzyme production, hemolysis, and bile salt hydrolase). Thus, we described our data as in vitro and in vivo results as major and the probiotic properties as relatively minor part in the text context. Thank you.

Point 4. It is not clear why the antimicrobial effect was evaluated in vitro using the supernatant of LAB cultures, when instead the experimentation on mice was carried out with lyophilized bacteria.

Response: Probiotics are live bacteria that have beneficial effects on the host. It is known that ingested lactic acid bacteria (LAB) settle in the intestine or vagina and produce useful components such as lactic acid. In order to evaluate the effect of the substances produced by the live LAB (probiotics) indirectly, the antibacterial activity was evaluated using the cell free supernatant (CFS) of LAB, and then the vaginitis relief effect was evaluated using the live lactic acid bacteria. And we notified this purpose in the sentence in revised MS (Line 233). Your comments helped make this study more convincing. Thank you.

Point 5. Furthermore, it would have been useful to include in paragraph 2.3 also the evaluation, at the end of the treatment, of the presence of the administered probiotics in the vaginal samples. This data would have confirmed the passage of bacterial strains in the vagina and their ability to adhere.

Response: We agree with your opinion. If we have the opportunity in further study, we would like to analyze presence/absence of the consumed LAB through NGS approach. Thank you for your comments.

Point 6. The opinion of the referee is that, despite the undoubted interest in the topic addressed by the authors, the work needs greater clarity in planning and goal setting.

Response: We revised our manuscript according to your comments above. Your comments were improved our data and helped make this study more convincing. Thank you.

Reviewer 2 Report

A very interesting study of the potential application of probiotics in bacterial vagnosis, which is a growing problem among women in the world. The paper is written clearly and concisely and the methodological experiments are well arranged. Can you specify in section 2.1. What standard media and conditions did you use to grow LAB?
Explain whether this is a strong inhibition of LAB on GV because the results do not show strong inhibition in vitro. Do you want to present the results as% inhibition or something else?
section 3.2. whether the result refers to cfu in vaginal fluid. It is not necessary to constantly list the full names of bacteria throughout the text. Prepare abbreviations in the prepositions.

Author Response

Point 1. Can you specify in section 2.1. What standard media and conditions did you use to grow LAB?

Response: We mentioned the culture medium (MRS broth) in the sentence as your comment. Thank you.

Point 2. Explain whether this is a strong inhibition of LAB on GV because the results do not show strong inhibition in vitro. Do you want to present the results as% inhibition or something else?

Response: We added GV inhibition rate under the cell count values. Your comments were greatly improved our data. Thank you.

Point 3. section 3.2. whether the result refers to cfu in vaginal fluid. It is not necessary to constantly list the full names of bacteria throughout the text. Prepare abbreviations in the prepositions.

Response: Yes. We counted the number of G. vaginalis from the vaginal wash fluid (Line 140-142). We changed the strain full names as abbreviation in the whole text as your comment. If there's something we haven't founded, please point it out next round. Thank you.

Round 2

Reviewer 1 Report

I thank the authors for the corrections made and for the replies to my comments. Unfortunately, however, the article still has many gaps. I still can't find the originality of the study, and the feeling of a title that doesn't reflect the real content of the article persists. Table 1 is inconsistent with the description in paragraph 2.2 of Materials and Methods section.  In this paragraph the authors declare that the number of G. vaginalis was measured at the baseline and after 36 hours of incubation while Table 1 reports the data at 12 and 24 hours. In the discussion section the authors added that LAB treatment inhibited the growth of G. vaginalis by up to 45% in one week. How did they calculate this percentage? I do not understand on what basis the authors state that, from this evidence it can be understood that the longer the LAB intake period, the greater the GV inhibition rate.

Author Response

I thank the authors for the corrections made and for the replies to my comments.

Response: Thank you very much for reviewing our paper despite your busy schedule.

We will actively reflect your review opinions and procced at further research.

Unfortunately, however, the article still has many gaps. I still can't find the originality of the study, and the feeling of a title that doesn't reflect the real content of the article persists.

Response: We correct the title as [In vivo confirmation of the antimicrobial effect of probiotic candidates against Gardnerella vaginalis] to represent the probiotic potential.

In the discussion section the authors added that LAB treatment inhibited the growth of G. vaginalis by up to 45% in one week. How did they calculate this percentage?

Response: We found some critical mistake the result of the inhibition rate according to your comments. Thus, we represented the calculation formula equation in M&M part in revised MS (Line 135-138), and we double checked our results throughout the text again and again.

I do not understand on what basis the authors state that, from this evidence it can be understood that the longer the LAB intake period, the greater the GV inhibition rate.

Response: We totally agree with your opinion. To avoid a logical leap, we deleted the sentence in revised MS. Your comments helped make this study more convincing. Thank you.